# Joint Learning Between Reference Image and Text Prompt for Fashion Image Editing

## Abstract

Fashion image editing is an essential tool for designers to visualize design concepts, aiming to modify the garment in an input fashion image while ensuring that other areas of the image remain unaffected. Existing methods primarily focus on images-based virtual try-on or text-driven fashion image editing, often relying on multiple auxiliary information including segmentation masks or dense poses. However, they struggle with error accumulation or high computational costs when performing try-on and editing simultaneously. In this work, we introduce a joint learning fashion image editing framework based on text prompts and reference images, named $D^2$-Edit. It aims at flexible, fine-grained editing including garment migration and attribute adjustments such as sleeve length, texture, color, and material via textual descriptions. Our proposed $D^2$-Edit consists of four key components: (i) **image degradation module**, which introduces controlled noise to facilitate the learning of the target garment concept and preserves the contextual relationships between the target concept and other elements; (ii) **image reconstruction module**, responsible for reconstructing both the fashion image and the reference image; (iii) **garment concept learning module** that encourages each text token (e.g., *skirt*) to attend solely to the image regions corresponding to the target concept via cross-attention loss; and (iv) **concept editing direction identification module**, designed to enable flexible attribute adjustments like fabric, color, and sleeve length. Extensive comparisons, ablations, and analyses demonstrate the effectiveness of our method across various test cases, highlighting its superiority over existing alternatives.

## 1 Introduction

Fashion image editing aims to modify an input fashion image to achieve enhanced or distinctive visual clothing effects, while enabling the adjustment of garment attributes such as color, texture, and fabric. This approach facilitates various applications for creating novel content, such as personalized outfit generation, virtual try-on experiences, and concept visualization. The advanced fashion image editing methods could satisfy a large variety of user requirements for modifying either a full image Baldrati et al. (2023); Song et al. (2023); Baldrati et al. (2024); Pernuš et al. (2025) or its local regions Huang et al. (2025); Wang & Ye (2024); Anonymous (2024).

Existing methods can be categorized into three groups: inpainting-based, product image-based and text-based fashion image editing methods, as illustrated in Fig. 1. The first two approaches Cui et al. (2024); Song et al. (2024a) typically rely on inputs such as source fashion images, reference images, and multi-modal cues (e.g., masks, keypoints) to perform virtual try-on. However, inpainting-based methods (see Fig. 1(a)) are generally limited to overlaying the garment from the reference image onto the source image, and their effectiveness heavily depends on the quality of the segmentation mask. Imperfect masks may result in misalignment or inaccuracies results. Product image-based methods (see Fig. 1(c)), on the other hand, rely on high-quality, background-free images of garments as reference inputs. While these images provide clear garment details, they are difficult to obtain in practical scenarios. Moreover, these methods often require multi-modal cues (e.g., segmentation masks, dense poses and keypoints), which are time-consuming and prone to errors in annotation. While both of these approaches can achieve virtual try-on, they lack the flexibility to edit garment attributes. In contrast, text-based fashion image editing methods (see Fig. 1(b)) allow users to guide edits through textual prompts, enabling adjustments to garment attributes like *color*, *style*, and *fabric*.

However, they struggle with more complex modifications (e.g., fabric texture or intricate details), as text descriptions often lack the precision needed for accurate virtual try-on results. For example, in Fig. 1(b), the details and style of the skirt are hard to describe accurately in text. Therefore, these methods fail to deliver accurate and realistic virtual try-on results.

Yet, all three types of existing methods fail to achieve the goal of dressing the person in the original image with the target garment while simultaneously editing other fashion attributes (e.g., *color*, *fabric*, *style*). To solve these issues, a straightforward way is to incorporate a virtual try-on method with a text-based image editing method (i.e., try-on first then edit, or edit first then try-on). However, this method suffers from the following drawbacks: 1) The two-phase editing process increases the processing time and reduces the overall efficiency. 2) Errors in the first phase may be amplified in the second phase, resulting in the final result deviation from expectations. 3) The accuracy of the

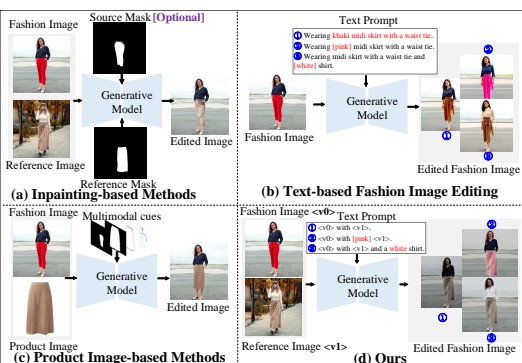

Figure 1: **Conceptual Comparisons for Different Pipelines.** (a) and (c) simulate virtual try-on using reference or product images but lack flexible editing capabilities; (b) support fine-grained fashion image editing but cannot achieve accurate virtual try-on results. (d) Our method leverages reference and original images with textual prompts to enable both precise virtual try-on and flexible garment edits (e.g., *texture*, *color*.)

editing result is highly dependent on the edit mask or other auxiliary information (e.g., keypoints, skeletons) provided by the user, which increases the burden on the user.

To address these issues, we propose a novel end-to-end framework, $D^2$**-Edit**, that allows for virtual try-on and garment attributes editing simultaneously via a joint learning of text prompt and reference images. Specifically, to ensure the relationships between the target garment concepts and other contextual elements are preserved while the target garment concepts are learned, we introduce an image degradation module (IDM). This module combines a pre-trained text-based image segmentation model with a weighted Gaussian noise degradation strategy. Then, the image reconstruction module (IRM) is proposed to reconstruct the fashion image and target garment in the reference images simultaneously. Next, to promote the learning of the desired garment concepts, we introduce the garment concept learning module (GCLM) to encourage each text token (e.g., *skirt, trousers*) to attend exclusively to the image regions occupied by the corresponding concept via cross-attention loss. Furthermore, we develop a clothing attribute editing direction identification module (CAEDIM) that uses a pair of text descriptions to determine the editing direction of a concept, enabling flexible editing of attributes such as *fabric* and *color*, etc. The main contributions are summarized as follows:

- We reveal three key challenges in fashion image editing method: a) inefficiency of the two-stage process, b) error accumulation, and c) reliance on user-provided masks.

- We propose a novel end-to-end framework, $D^2$**-Edit**, which contains four modules: i) IDM to learn the target garment and preserve its contextual relationship with other elements, ii) IRM to ensure accurate image reconstruction, iii) GCLM to encourage the alignment of the target garment concept with text tokens, and iv) CAEDIM to enable controllable attribute adjustments, e.g., *color*, *texture*.

- Comprehensive qualitative and quantitative experiments validate the effectiveness of $D^2$-Edit, demonstrating its superiority over other state-of-the-art methods.

## 2 METHODOLOGY

### 2.1 PROBLEM SETUP

In this study, Let $I, I^r \in \mathbb{R}^{H \times W}$ represent the input fashion image and the reference image, respectively, where $H$ and $W$ denote the height and width of the image, respectively. $I^r$ provides the target garment concept for modification. The editing process is guided by a text prompt $P$, which specifies the desired modifications (e.g., *clothing type, color, fabric*). Additionally, auxiliary prompt pair $(t_0, t_1)$ is introduced to encode the directions of conceptual transformations. The goal of fashion

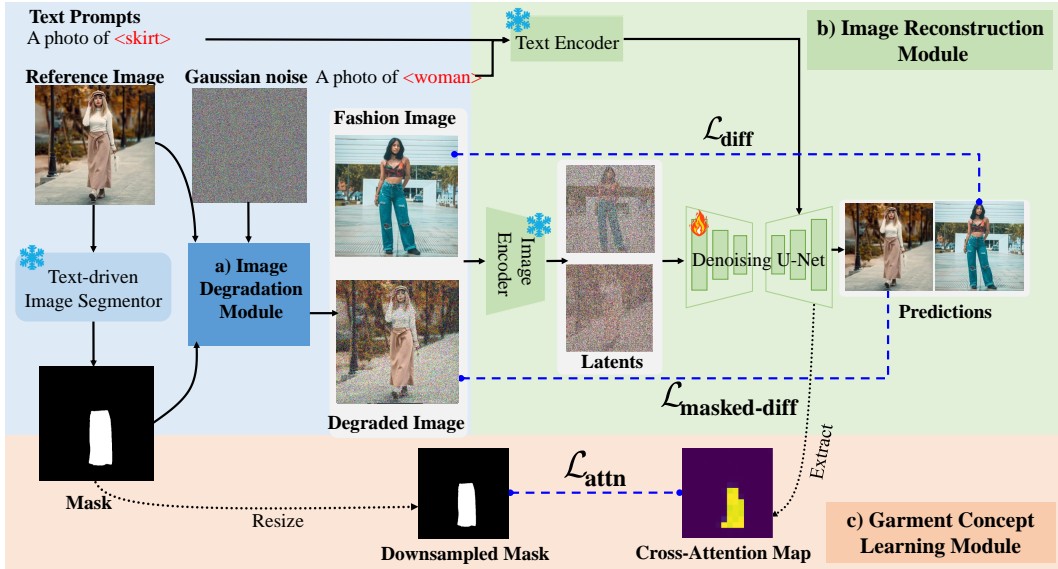

Figure 2: **The Overview of D²-Edit.** Our D²-Edit consists of four key components: (a) Image degradation module. (b) Image reconstruction module. (c) Garment concept learning module. (d) Clothing attribute editing direction identification module. Module (d) is not shown in this figure.

image editing is to identify a function $\mathcal{F}$ that generates the edited image $I' = \mathcal{F}(I, P, I^r, (t_0, t_1))$, modifying specific garment attributes in $I$ according to $P$, while preserving the unaffected regions.

## 2.2 OVERVIEW

Fig. 2 illustrates the framework of our D²-Edit, which aims to generate a new fashion image $I'$ by modifying specific conceptual attributes (e.g., *clothing type, color*) of the input fashion image $I$, while leaving the unrelated region unchanged. The specific implementation of D²-Edit consists of four steps: 1) **Image Degradation Module** (§ **2.3**) is designed to disturb irrelevant visual semantics via a weighted Gaussian noise degradation strategy, and thus suppresses the model's sensitivity to irrelevant visual details while preserving the overall visual context; 2) **Image Reconstruction Module** (§ **2.4**) is proposed to ensure simultaneous learning of both the fashion image $I$ and the target garment concept in the reference image $I^r$; 3) **Garment Concept Learning Module** (§ **2.5**) is employed to strengthen the correlation between the learned target garment concept and text token $v$; and 4) **Attribute Editing Direction Identification** (§ **2.6**), which enables flexible text-driven fashion image editing by mapping a predefined text prompt pair $(t_0, t_1)$ (e.g., "long-sleeved shirt" and "shirt" for *sleeve length* attribute.) into the denoising U-Net representation space and computing their vector difference to determine semantic editing directions for specific attributes.

## 2.3 IMAGE DEGRADATION MODULE

To focus on the target garment concepts in the reference image while preserving their contextual relationships with other elements, the image degradation module is introduced to obtain the degraded reference image. Specifically, we first utilize a pre-trained text-guided semantic segmentation model, i.e., Grounded-SAM Ren et al. (2024), to extract the semantic segmentation mask $M_c = \text{Grounded-SAM}(I^r, P_r)$ corresponding to the garment concepts (e.g., *shirt*) in the reference image. Here, $P_r$ is the text description for the target garment in reference image $I^r$. This segmentation mask $M_c$ serves two key purposes: a) To ensure the learned garment concepts can seamlessly integrate into the original fashion image, we combine the obtained mask with a weighted Gaussian image degradation strategy to perturb unrelated regions of the reference image. This strategy preserves the contextual relationships between the target garment concepts and the rest of the image, while perturbing irrelevant regions to minimize the influence of extraneous visual information. Specifically, we first generate a Gaussian noise matrix $N^r \sim \mathcal{N}(0, 1)$ with the same shape as the reference image $I^r$. Then, we combine the noise matrix $N^r$ with the corresponding mask $M_c$ and

apply it to the reference image $I^r$. The degradation process is:

$$I^{rd} = \alpha N^r \odot (1 - M_c) + I^r, \tag{1}$$

where $I^{rd}$ is the degraded reference image, $\alpha$ is a weight controlling the noise intensity, and $\odot$ denotes element-wise multiplication. $M_c$ is the mask indicating the target garment region. b) In the GCLM module, the extracted mask $M_c$ is further employed to compute the difference with the cross-attention maps extracted from the denoising U-Net. This process encourages each text token $v$ to attend exclusively to the image regions occupied by the corresponding garment. More detailed description will be given in § 2.5.

## 2.4 IMAGE RECONSTRUCTION MODULE

Further, IRM is designed to reconstruct both the original fashion image and the target garment concept in the reference image accurately. Specifically, on one hand, to ensure faithful reconstruction of the original fashion image in pixel space, we employ a traditional diffusion loss, i.e.,

$$\mathcal{L}_{\text{diff}} = \mathbb{E}_{(I_t, e, t)} \left[ ||\epsilon(I_t, e, t) - \varepsilon||_2^2 \right]. \tag{2}$$

Here, $\varepsilon \sim \mathcal{N}(0, 1)$ is the unscaled noise, and $\epsilon(\cdot)$ represents the denoising U-Net, $I_t = \sqrt{\alpha_t} I_0 + \sqrt{(1 - \alpha_t)}\epsilon$ is the noisy latent image of $I$ at the $t$-th time step during the diffusion process, where $\alpha_t$ denotes a predefined variance schedule, latents $I_0 = E_I(I)$ is obtained by image encoder $E_I$. The text embedding $e = E_T(P_s)$ is obtained by encoding the source prompt $P_s$ with the text encoder $E_T(\cdot)$. On the other hand, unlike the learning objective for the original image, our goal is to focus on learning the target garment concepts in the reference image rather than the entire image. To achieve this, we use a masked diffusion loss that encourages the model to accurately reconstruct the target garment concepts. This can be formalized as follows:

$$\mathcal{L}_{\text{masked-diff}} = \mathbb{E}_{(I_t^r, e_r, t)} \left[ ||M_c' \odot \epsilon(I_t^r, e_r, t) - \varepsilon_r||_2^2 \right], \tag{3}$$

where $M_c'$ is resized from $M_c$ to match the shape of unscaled noise $\varepsilon_r$ and $I_t^r$, $e_r = E_T(P_r)$ is the text embedding of the source textual prompt $P_r$, such as "*a photo of $<v>$.*" .

## 2.5 GARMENT CONCEPT LEARNING MODULE

To acquire target garment concepts from the reference image, the GCLM module is designed to strengthen the correlation between the visual semantics of garment concepts and their corresponding text tokens $v$, where cross-attention loss $\mathcal{L}_{\text{att}}$ is proposed to enhance the model's focus on desired garment concepts by computes the discrepancy between the cross-attention map $CA(v, I_t^r)$ (associated with the text tokens $v$) extracted from the denoising U-Net and the corresponding segmentation mask. Specifically, during the model fine-tuning phase, we first extract attention maps $CA(v, I_t^r)$ from the denoising U-Net that correspond to the newly added text tokens $v$ with a resolution of $16 \times 16$, which contain the most semantic information Hertz et al. (2022). These maps are then normalized to the range $[0, 1]$, and the difference between them and the resized mask $resize(M_c)$ is computed as the cross-attention loss $\mathcal{L}_{\text{att}}$. This procedure can be written as Eq. (4).

$$\mathcal{L}_{\text{att}} = \mathbb{E}_{(I_t^r, t)} [||CA(v, I_t^r) - resize(M_c)||_2^2], \tag{4}$$

where $CA(v, I_t^r)$ denotes cross-attention maps between text token $v$ and latent noisy image $I_t^r$ at the $t$-th time step averaged over the cross-attention layers of the upsampling blocks in the denoising U-Net model, $resize(M_c)$ represents the resized version of the generated concept mask $M_c$ that matches the shapes of the cross-attention maps.

Finally, the overall loss of $D^2$-Edit can be formalized as

$$\mathcal{L} = \lambda_{\text{diff}} \mathcal{L}_{\text{diff}} + \lambda_{\text{mask}} \mathcal{L}_{\text{masked-diff}} + \lambda_{\text{att}} \mathcal{L}_{\text{att}}. \tag{5}$$

Here, $\lambda_{\text{diff}}, \lambda_{\text{mask}}, \lambda_{\text{att}}$ are empirically set to 1, 1, and 2e$-$2,, respectively. The first two terms of $\mathcal{L}$ ensure that the visual semantics of garment concepts and fashion images are learned simultaneously, while the last term enhances the correlation between the semantics of garment concepts and the corresponding text token $v$. Note that to enable efficient fine-tuning, we train only the denoising U-Net $\epsilon$ using LoRA and the embedding vector of the text token $v$ while freezing the other components.

---

**Algorithm 1** The overall process of D$^2$-Edit.

---

**Require**: Fashion Image $I$, Reference Image $I^r$, Source Prompt $P_s$, Target Prompt $P$, Reference Prompt $P_r$, Auxiliary Prompts $t_0, t_1$, Degradation weight $\alpha$; Pre-trained CLIP Text Encoder $E_T$, Image Encoder $E_I$, Denosing U-Net $\epsilon$, Guidance scale $w$, Concept intensity weight $\gamma$.
**Output**: Well-trained Model $\{\epsilon, E_T, E_I, D\}$.

1: $M_c = \text{Grounded-SAM}(I^r, P_r)$. # Get segmentation mask of target garment concept.
2: # (Training Phase).
3: **for** training step $t$ in $[0, T]$ **do**
4:     # Image degradation.
5:     $I^{rd} = \alpha N^r \odot (1 - M_c) + I^r$, $N^r \sim N(0, 1)$, # Get the degraded image.
6:     $(I_0, I_0^r) = E_I((I, I^{rd}))$, # Convert image to latents.
7:     $(I_t, I_t^r) = \sqrt{\alpha_t}(I_0, I_0^r) + \sqrt{(1 - \alpha_t)}\epsilon$ #Noisy latents
8:     $(e, e_r) = E_T((P_s, P_r))$, # Get the text embedding.
9:     $(\epsilon', \epsilon_r') = \epsilon((I_t, I_t^r), (e, e_r), t)$ ,# Predict the noise residual.
10:     Optimize $\epsilon$ via Eq. (5).
11: **end for**
12: # (Inference Phase).
13: $(e_{\text{t}}, e, e_\emptyset) = E_T(P), E_T(P_s), E_T(\text{" "})$, # Get the text embedding.
14: $e_0, e_1 = E_T(t_0), E_T(t_1)$, # Get the text embedding.
15: **for** inference step $t$ in $[0, T]$ **do**
16:     $s_\emptyset = \epsilon(I_t, e_\emptyset, t)$ ,# Unconditional score.
17:     $s_{\text{src}}, s_{\text{tar}} = \epsilon(I_t, e, t), \epsilon(I_t, e_{\text{t}}, t)$, # Conditional score.
18:     Compute the clothing attribute editing direction $\Delta C_t$ via Eq. (7).
19:     Compute new conditional score via Eq. (6).
20:     $s \leftarrow s_\emptyset + w(s_{\text{new}} - s_\emptyset)$. # For attribute editing.
21:     $I_{t-1} \sim \mathcal{N}(I_t - s), \sigma_t^2 I)$.
22: **end for**

---

## 2.6 ATTRIBUTE EDITING DIRECTION IDENTIFICATION

Now, our method leverages text prompts to integrate garment concepts learned from reference images into fashion images, but it does not support fine-grained clothing attribute editing, such as *fabric texture, color, sleeve length*, or *skirt length*. To overcome this limitation, we introduce a novel module in the inference phase, inspired by Wang et al. (2024), which constructs an attribute editing space using only a pair of auxiliary textual descriptions. By identifying the clothing attribute editing direction $\Delta C_t$, our method generates new conditional score $s_{\text{new}}$ to enable flexible fashion image editing. This process is formally expressed as:

$$s_{\text{new}} = s_{\text{src}} - \lambda \cdot \langle (s_{\text{tar}} - s_{\text{src}}), \Delta C_t \rangle \cdot \Delta C_t \tag{6}$$

Here, $s_{\text{src}} = \epsilon(I_t, e, t)$ and $s_{tar} = \epsilon(I_t, e_t, t)$, where $e_{\text{t}}$ represents the text embedding of the text prompt $P$ used for target attribute editing. $\langle (s_{\text{tar}} - s_{\text{src}}), \Delta C_t \rangle$ denotes the inner product between the difference vector and the attribute direction vector. The term $s_{\text{tar}} - s_{\text{src}}$ denotes the score difference. The attribute editing direction vector $\Delta C_t$ at time step $t$ controls the editing process of a specific concept in the fashion image, and the weight $\gamma$ modulates the effect of that direction, $w$ represents the guidance scale that controls the balance between text adherence and image diversity. By projecting the score difference onto the attribute editing direction $\Delta C_t$, the image editing is guided along the desired direction, preventing unintended changes to other areas of the image. Specifically, we provide a pair of text prompts $t_0$ and $t_1$ like *"a long-sleeved shirt"* and *"a short-sleeved shirt"* to define an attribute editing direction $\Delta C_t$ by

$$\Delta C_t = \frac{\epsilon(I_t, e_0, t) - \epsilon(I_t, e_1, t)}{||\epsilon(I_t, e_0, t) - \epsilon(I_t, e_1, t)||_2}, \tag{7}$$

where $e_0$ and $e_1$ are the embedding of text prompts $t_0$ and $t_1$, with $e_0 \neq e_1$. See **Appendix** ( § **B.5**) for $e_0 = e_1$ case. The overall process is summarized in Algorithm 1.

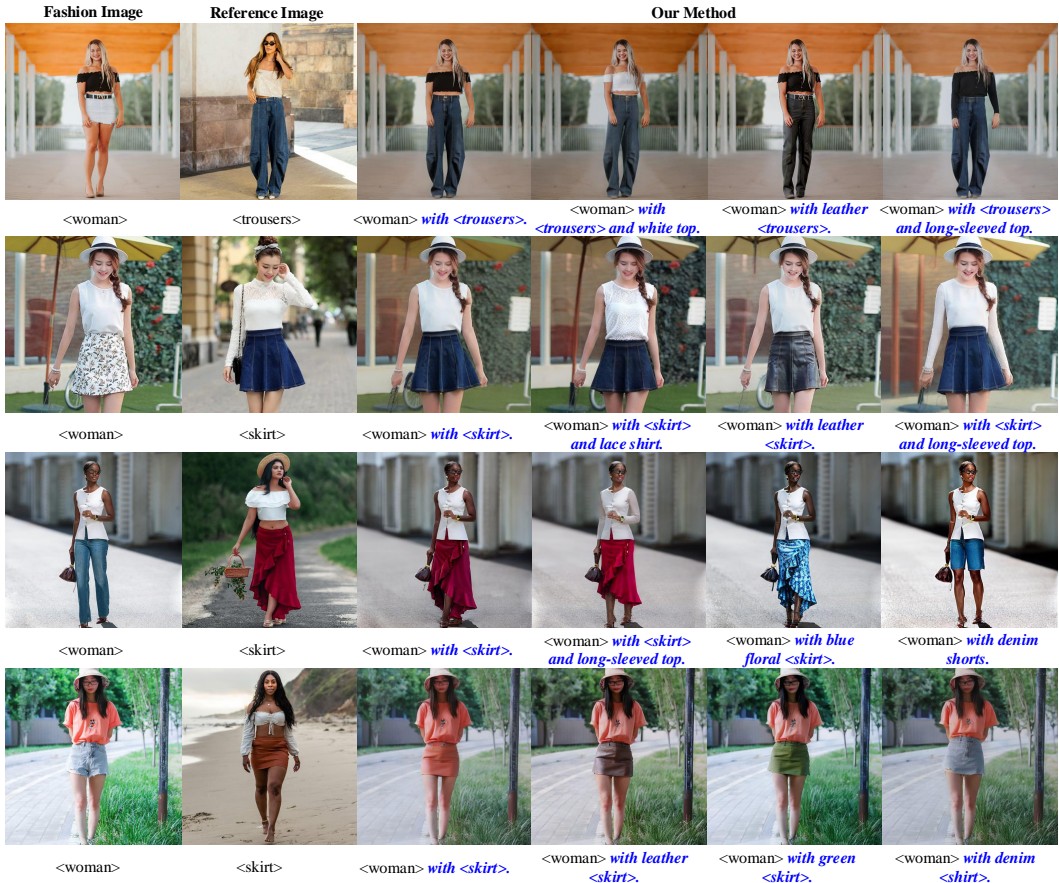

Figure 3: **Experimental Results of Our Method.** By providing a reference image and the original fashion image, our method enables fashion image editing based on the reference image through the target editing text prompt, and simultaneously enables editing of multiple clothing attributes, including texture, color, material, and sleeve length. More results are shown in **Appendix § B**.

## 3 EXPERIMENTS

### 3.1 IMPLEMENTATION DETAILS

**Setup.** Following the prior work Song et al. (2024b); Zhou et al. (2024), we employ the official pre-trained Stable Diffusion v2.1-base Rombach et al. (2022) as our foundational model, downsampling all images to a resolution of $512 \times 512$ pixels for consistency across experiments, and the LoRA rank is set to 512. In addition, we set the training steps to 1000 and learning rate to 1e-4, using AdamW Loshchilov (2017) as the optimizer. The degradation weight $\alpha$ is experimentally determined to be 1. The guidance scale is empirically set to 5, balancing text adherence and image diversity. All experiments are conducted on one NVIDIA RTX 4090 GPU with 24 GB of memory.

**Dataset.** To facilitate a fair result comparison with state-of-the-art methods, we follow the settings in Cui et al. (2024); Song et al. (2024b) and conduct experiments on two commonly used datasets, including the StreetTryOn Dataset Cui et al. (2024) and the fashion image dataset from Unsplash Unsplash (2025). The StreetTryOn Dataset Cui et al. (2024) is a fashion image dataset specifically designed for virtual try-on tasks, derived from the large-scale fashion retrieval dataset DeepFashion2 Ge et al. (2019). It comprises 12,364 street fashion images for training and 2,089 for validation. Additionally, following the prior work Huang et al. (2025); Song et al. (2024b); Zhang et al. (2023); Kawar et al. (2023), we select multiple fashion street photography images from Unsplash, encompassing both full-body and half-body styles.

**Baselines.** To verify the effectiveness of $D^2$-Edit, we compare it with the following baselines: Break-A-Scene Avrahami et al. (2023) & blended diffusion Avrahami et al. (2022) (BAS-BD), Any-

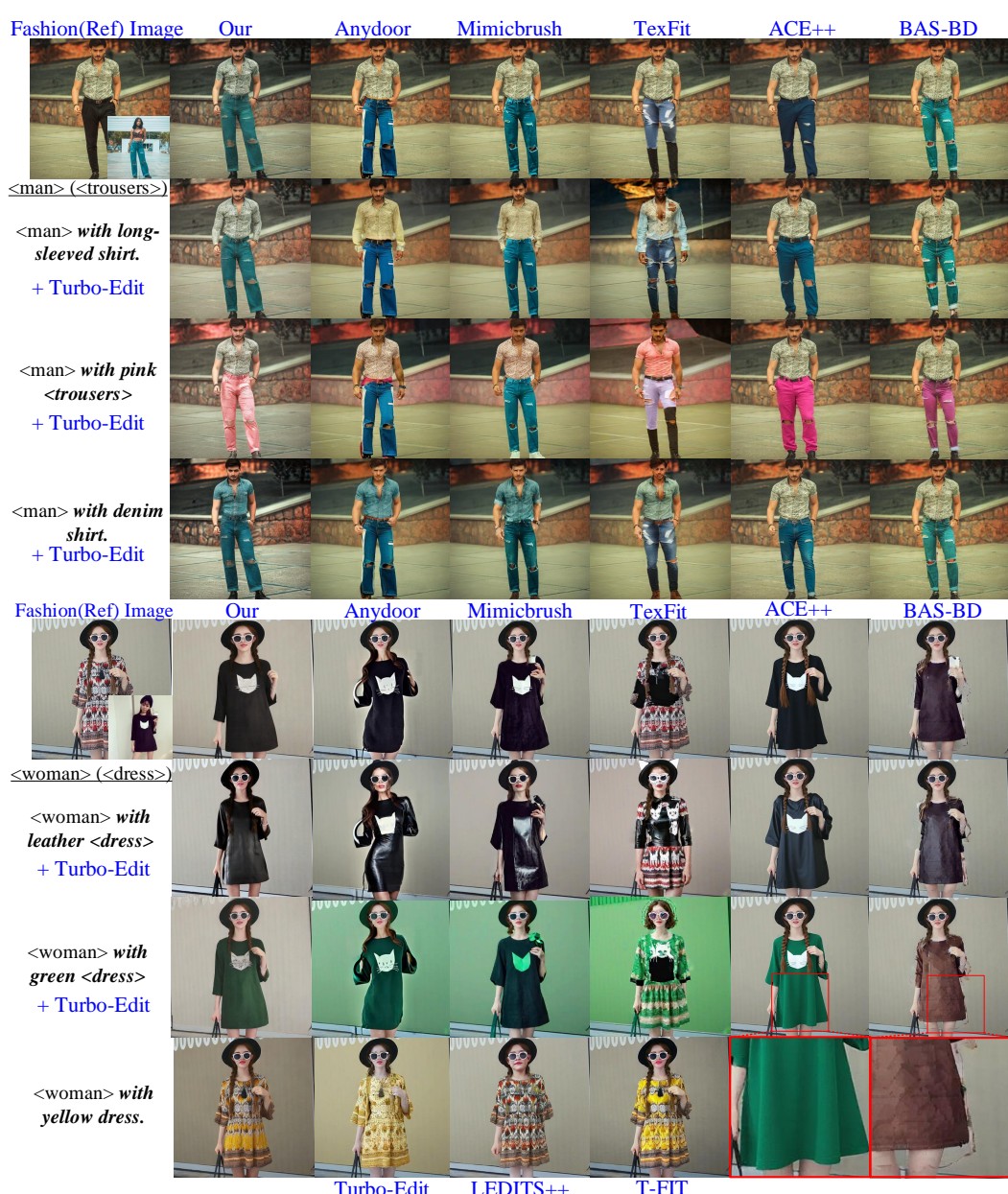

Figure 4: **Qualitative comparison.** Given the input person image and the reference garment (first column), we compare editing results across different methods under various text instructions (rows). Columns 2–7 show different methods, with Columns 3 and 4 requiring Turbo-Edit for editing (otherwise limited to try-on). Edited images highlighted in red box show noticeable artifacts.

door Chen et al. (2024), Mimicbrush Zhao (2024), TexFit Wang & Ye (2024), and ACE++ Mao et al. (2025). Note that these four methods are combined with advanced text-driven editing methods (i.e., Turbo-Edit Deutch et al. (2024), LEDITS++ Brack et al. (2024), and T-FIT Huang et al. (2025)) to achieve more diverse editing results. More details of these methods are provided in **Appendix** § **C**.

 **Evaluation Metrics.** We evaluate the similarity between the edited image and the original fashion image by calculating the learned perceptual image patch similarity (LPIPS) Zhang et al. (2018) and peak signal-to-noise ratio (PSNR) Wang et al. (2004). Note that, LPIPS* and PSNR* are used to calculate the similarity between the edited image and the target garment in the reference image. Specifically, we first extract the target garment regions from both the edited and reference images, and then calculate the PSNR and LPIPS between these two regions. This allows for a quantitative assessment of the visual consistency between the garment in the edited image and the garment in

Table 1: **Quantitative Comparisons.** Optimal results in bold, suboptimal results underlined. †
denotes the results obtained by combining these methods with the text-driven editing method, Turbo-
edit. M and V denote mean and variance, respectively.

| Method | LPIPS (M ± V)↓ | PSNR (M ± V)↑ | CLIP-T (M ± V)↑ | CLIP-I (M ± V)↑ | KID (M ± V)↓ |
|---|---|---|---|---|---|
| **Our** | 0.2417 (± 0.032) | 29.37 (± 0.487) | **0.1656 (± 0.056)** | **0.9386 (± 0.020)** | **0.0265 (± 0.013)** |
| **Mimicbrush†(NeurIPS, 2024)** | 0.2760 (± 0.035) | 29.29 (± 0.836) | 0.1528 (± 0.023) | 0.9378 (± 0.016) | 0.0551 (± 0.015) |
| **Anydoor† (CVPR, 2024)** | 0.2705 (± 0.060) | 30.00 (± 1.328) | 0.1574 (± 0.021) | 0.9189 (± 0.045) | 0.0344 (± 0.015) |
| **BAS-BD (SIGGRAPH, 2023)** | 0.2545 (± 0.003) | **30.36 (± 0.352)** | 0.1546 (± 0.001) | 0.8966 (± 0.002) | 0.0459 (± 0.008) |
| **TexFit (AAAI, 2024)** | 0.2944 (± 0.003) | 20.97 (± 54.65) | 0.1245 (± 0.002) | 0.8901 (± 0.003) | 0.0299 (± 0.002) |
| **ACE++ (Arxiv, 2025)** | **0.1964** (± 0.005) | 15.59 (± 4.570) | 0.1254 (± 0.001) | 0.8607 (± 0.002) | 0.0758 (± 0.007) |

Table 2: **Quantitative Comparisons.** Optimal results in bold, suboptimal results underlined. *
Indicates that for each metric, we focus on the image similarity between the target garment in the
reference image and the corresponding garment in the edited image. "Time" represents the average
time spent during the inference process. M and V denote mean and variance, respectively.

| Method | LPIPS* (M ± V)↓ | PSNR* (M ± V)↑ | CLIP-I* (M ± V)↑ | Time↓ |
|---|---|---|---|---|
| **Our** | **0.1479** (± 0.002) | **19.10** (± 0.573) | **0.9132** (± 0.000) | **6s** |
| **Mimicbrush (NeurIPS, 2024)** | 0.1928 (± 0.002) | 18.22 (± 0.149) | 0.9049 (± 0.000) | 31s |
| **Anydoor (CVPR, 2024)** | 0.2035 (± 0.004) | 16.98 (± 2.247) | 0.8796 (± 0.003) | 40s |
| **BAS-BD (SIGGRAPH, 2023)** | 0.1686 (± 0.003) | 19.04 (± 0.978) | 0.8302 (± 0.004) | 12s |
| **TexFit (AAAI, 2024)** | 0.2228( ± 0.009) | 16.22 (± 14.54) | 0.8026 (± 0.005) | 24s |
| **ACE++ (Arxiv, 2025)** | 0.2253 (± 0.009) | 13.64 (± 11.08) | 0.8385 (± 0.005) | 21s |

the reference image. Further, CLIP-based metrics Radford et al. (2021) are employed to assess two
aspects: **image-text alignment**, by computing the CLIP similarity between the edited fashion im-
ages and the target text prompts (i.e., CLIP-T), and **identity preservation**, by measuring the cosine
similarity between the edited image and the original fashion image using CLIP image embeddings
(i.e., CLIP-I). We also report Kernel Inception Distance (KID) Bińkowski et al. (2018) to assess the
fidelity of the edited fashion images.

## 3.2 COMPARISON WITH STATE-OF-THE-ART METHODS

In this section, we present a comprehensive comparison of the editing results achieved by our $D^2$-
Edit against other state-of-the-art methods, both quantitatively and qualitatively. It is worth noting
that since Anydoor Chen et al. (2024) and Mimicbrush Zhao (2024) are only capable of virtual try-on
but fail to achieve other attribute editing simultaneously, for this reason, we select three state-of-the-
art text-driven image editing methods (i.e., Turbo-Edit Deutch et al. (2024), LEDITS++ Brack et al.
(2024), and T-FIT Huang et al. (2025)) for clothing attribute editing. Due to page limitations, addi-
tional results of LEDITS++, Turbo-Edit, and T-FIT are provided in the **Appendix** § **C.2**.

**Qualitative Comparison:** Fig. 3 illustrates the experimental results of our method and Fig. 4 pro-
vides a comparative analysis against other baselines. From Fig. 4, it is evident that the edited images
obtained by the AnyDoor, TexFit, ACE++, and BAS-BD often exhibit noticeable artifacts, which can
be attributed to their heavy reliance on semantic segmentation masks for the edited regions. This
dependency makes it challenging to seamlessly integrate the target garment concepts with the origi-
nal fashion images. Additionally, AnyDoor, MimicBrush, and TexFit struggle to edit other clothing
attributes simultaneously. To address this, we combined them with methods like Turbo-Edit, and the
experimental results are shown in rows 2-4 and 6-7 of Fig. 4. We observe that the editing results
either alter irrelevant areas(e.g., body identity or background), or deviate from the desired editing
target due to errors in the first phase. Furthermore, ACE++ often preserves the source garment's
shape or pleats (Fig. 4, col 6), which contradicts the goal of faithful editing—these should instead
conform to the target garment in reference image. In contrast, our approach not only enables the
seamless try-on of a specific garment item, but also enables the editing of various clothing attributes
simultaneously, including the *fabric, color, sleeve length*, etc. This is largely due to our method's
joint learning of fashion images and reference images and CAEDIM.

**Quantitative Comparison:** Tab. 1 show that our method overall outperforms other methods in
terms of LPIPS, PSNR, CLIP-I, and KID, indicating that the edited images of our $D^2$-Edit perform
better in terms of background preservation. This is mainly attributed to the IRM module, which
allows the model to better reconstruct the detailed information of the original fashion image. In
addition, our method also achieves higher scores on the CLIP-T, indicating that $D^2$-Edit can follow
the text editing instructions more accurately and generate editing results that meet expectations. Al-
though our LPIPS is slightly higher than ACE++ and PSNR marginally lower than BAS-BD, these

metrics are misleading: ACE++ erroneously preserves the original garment's shape or pleats, while these two methods cannot edit beyond the mask—e.g., the *fabric* and *sleeve length* of 'shirt' in Fig. 4 remains unchanged. Further, Tab. 2 shows the similarity between the tried-on garment generated by our method and the target garment in reference images. The results illustrate that our method is more effective in learning the target garment and achieving more accurate garment try-on. Morerover, $D^2$-Edit consistently outperforms both one-stage and two-stage approaches in terms of inference time. Overall, our method not only achieves precise fashion image editing but also keeps unrelated areas unchanged, outperforming existing methods in both objective and subjective metrics.

### 3.3 ABLATION STUDY

We validate the effectiveness of $D^2$-Edit by individually removing key modules. The experiment results are shown in Fig. 5, with additional quantitative comparisons in the **Appendix** § **E**. Red boxes highlight noteworthy aspects of the edited results produced by the various variants of $D^2$-Edit.

**Effect of the IDM.** From Fig. 5, we can observe that removing IDM leads to the model inadvertently learning irrelevant garment concepts, resulting in unintended alterations to non-target regions in the edited images. For instance, as shown in Fig. 5, row 2, col 2 & 4, the person's position and background in the edited image are incorrectly influenced by the reference image. This result suggests that IDM effectively prevents the model from being influenced by irrelevant regions by perturbing these areas, thereby ensuring the learning process focuses on the desired garment concepts.

**Effect of the $\mathcal{L}_{\text{masked-diff}}$.** Experimental results show that even with IRM, the absence of the masked diffusion loss $\mathcal{L}_{\text{masked-diff}}$ in our $D^2$-Edit leads to unintended influences from the reference image. As depicted in Fig. 5, the edited images exhibit undesired elements from the reference image, such as the person's position and body shape. This underscores the effectiveness of $\mathcal{L}_{\text{masked-diff}}$ in accurately reconstructing the target garment elements from the reference image, ensuring that only the intended garment concept is transferred during the editing process.

**Effect of the CAEDIM.** As illustrated in Fig. 5, the removal of CAEDIM significantly affects the model's editability. The edited image generated by $D^2$-Edit without CAEDIM is limited by the original fashion style of the image, making it difficult to achieve large structural edits, such as changing the *sleeve length* to long sleeves. In contrast, $D^2$-Edit with CAEDIM can produce more diverse editing results by progressively identifying the clothing attribute editing directions based on differences in predicted noise within the representation space. The sensitivity analysis of the degradation weight $\alpha$ and concept intensity weight $\gamma$ is provided in **Appendix** § **D**. Overall, the extensive ablation study indicates the effectiveness of each key component in $D^2$-Edit.

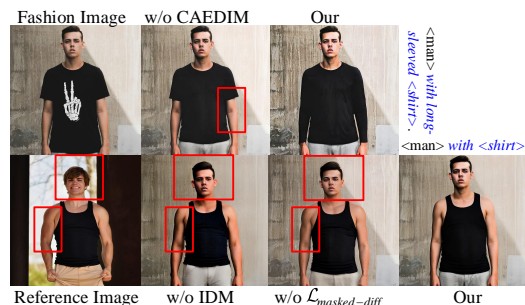

Figure 5: **Qualitative ablation studies.**

## 4 CONCLUSION

In this paper, we introduce a novel fashion image editing method driven by both text and reference images, addressing the limitations of existing methods that rely on user-supplied edit masks and employ a two-stage framework leading to error accumulation and inefficiency. By introducing four key modules—the image degradation module, image reconstruction module, garment concept learning module, and clothing attribute editing direction identification module—our method effectively learns garment concepts while preserving contextual relationships and ensuring precise image reconstruction. Additionally, the identification of editing directions in representation space enables more diverse editing outcomes. Experimental results demonstrate that our method achieves end-to-end fashion image editing and virtual try-on using only a text prompt, an original fashion image, and a reference image, eliminating the need for manual editing masks or the two-stage process. However, our method currently faces limitations in that it lacks an explicit module to preserve the brightness of the original fashion image. In future work, we plan to incorporate perceptual color spaces, like CIELAB, to overcome this limitation.

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

## A  STATEMENT ON LLMS USAGE

Large Language Model (LLMs) was used solely for language polishing. It did not contribute to the research design, analysis, or conclusions, which remain the sole responsibility of the authors.

## B  MORE EXPERIMENTAL RESULTS

In this section, we demonstrate the performance of our proposed method under various conditions, including its compatibility with different versions of Stable Diffusion (SD), its effectiveness when the reference images are product flat-lay photos, its capability in handling complex and fine-grained fashion attribute editing tasks, and its robustness in scenarios where there is a significant pose difference between the source and reference images. Additionally, we investigate the impact of different settings for $e_0$ and $e_1$ in Eq. (7), and present additional editing results produced by our $D^2$-Edit framework. Fig. 6-11 show the results of our method under various editing scenarios.

### B.1  COMPATIBILITY ACROSS DIFFERENT SD VERSIONS

To evaluate the performance of our method across different SD versions, we test $D^2$-Edit on two widely used models: SD v1.5 and SD 2-1-base Rombach et al. (2022). As shown in Fig. 6, while there are slight variations in the visual results across versions, our method consistently demonstrates comparable fashion editing capabilities. This indicates that $D^2$-Edit is generalizable and compatible with different versions of Stable Diffusion.

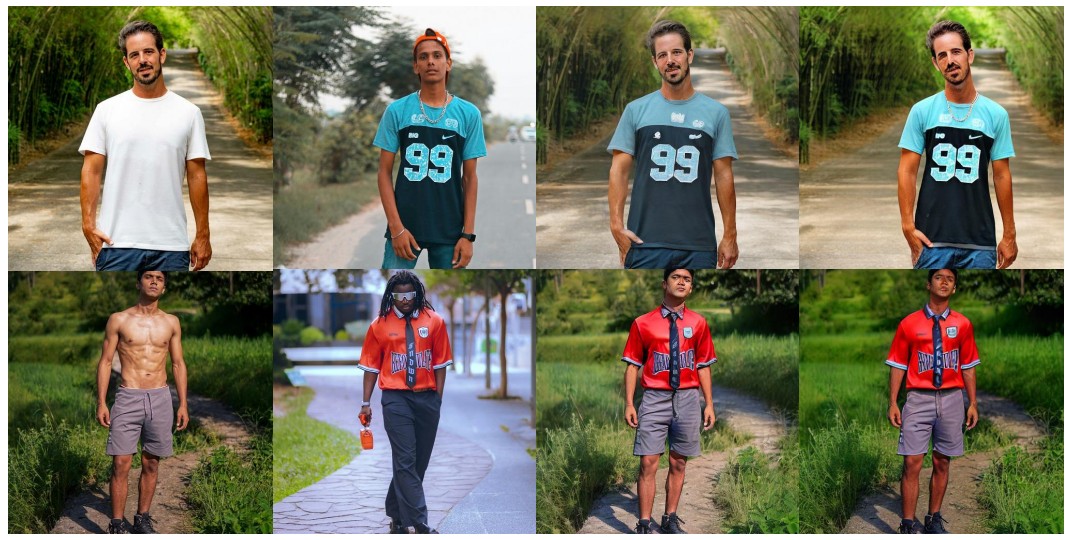

**Source Image      Reference Image      Our(SD-2-1-base)      Our(SD-v1-5)**

Figure 6: Experimental results with different SD versions.

### B.2  PERFORMANCE ON PRODUCT FLAT-LAY IMAGES

Fig. 7 presents try-on results generated by our method using flat product images as reference. As shown, our method still achieves satisfactory editing performance under this setting, demonstrating its adaptability. It is important to highlight that our method is primarily designed to work with in-the-wild reference images—i.e., images of people wearing the target garments—due to the practical difficulty of obtaining clean, flat product images.

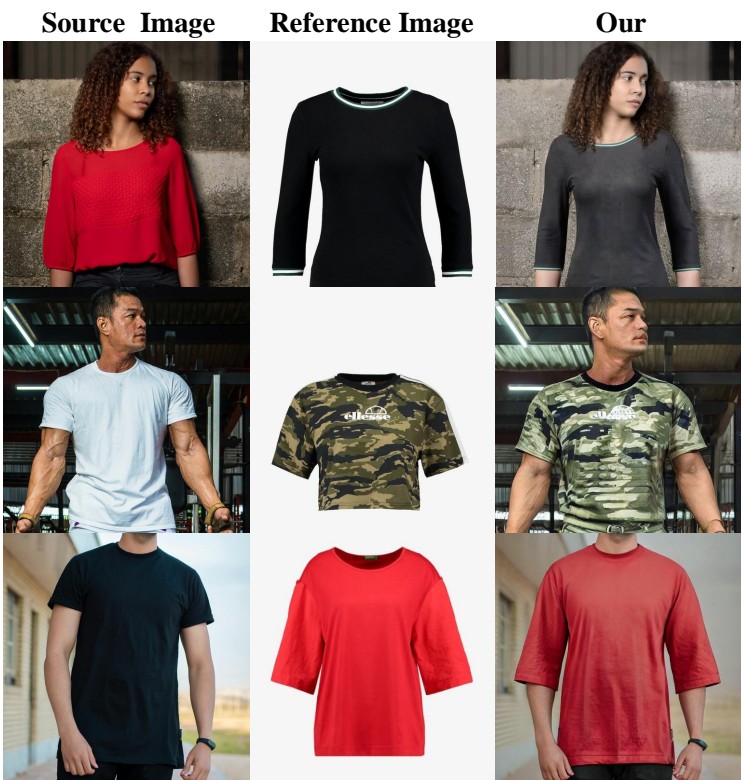

Figure 7: Effectiveness of D²-Edit with Clean Product Garment References.

## B.3 PERFORMANCE ON COMPLEX AND FINE-GRAINED EDITS

Fig. 8 illustrates the results of our method on complex, fine-grained, and multi-attribute editing tasks, where different colored texts indicate distinct fashion attributes. As shown, our method generalizes well to challenging scenarios involving intricate and diverse attribute manipulations, further demonstrating its effectiveness in handling complex fashion editing tasks.

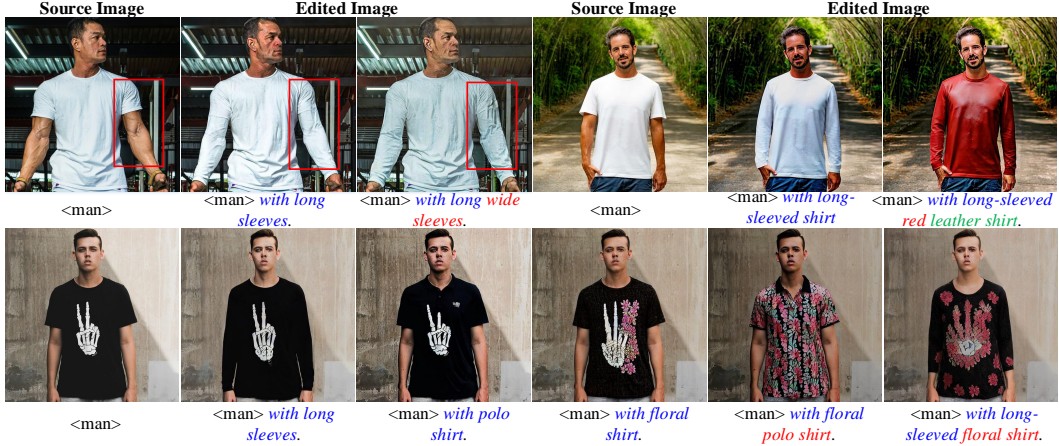

Figure 8: Experimental results of complex editing tasks. Different colored texts indicate distinct fashion attributes.

## B.4 ROBUSTNESS TO LARGE POSE DIFFERENCES

We investigate the performance of our method under scenarios where the reference image and the original fashion image exhibit significant pose differences. As shown in Fig. 9, our method still achieves favorable editing results despite the challenging discrepancies in human pose, further demonstrating its robustness and generalizability.

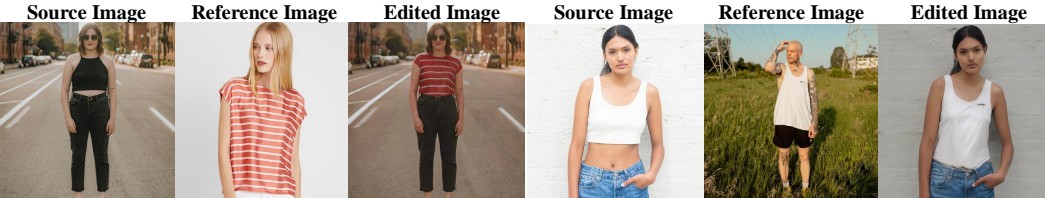

Figure 9: Experimental results under different poses/viewpoints.

## B.5 IMPACT OF DIFFERENT SETTINGS FOR $e_0$ AND $e_1$ IN EQ. (7)

To further validate the impact of different settings for $e_0$ and $e_1$ in Eq. (7), we explore two scenarios: (1) $e_0 = e_1$ and (2) $e_0 \neq e_1$, and evaluate their performance on both try-on and editing tasks. The results are presented in Fig. 10. As shown, when $e_0 = e_1$, the model is able to perform garment try-on but fails to support attribute editing. In contrast, when $e_0 \neq e_1$, our method can not only preserve the try-on capability but also enable controllable fashion attribute editing. Therefore, in our final design, we adopt the $e_0 \neq e_1$ setting by providing distinct prompt texts to specify user-desired garment attributes.

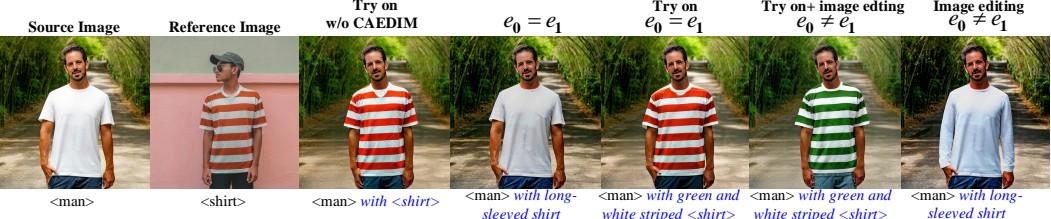

Figure 10: More results of our method. Our method enables garment try-on, original fashion image editing, and the simultaneous editing of try-on images. Blue colored texts indicate added or modified garment concepts.

## B.6 MORE EXPERIMENTAL RESULTS OF $D^2$-EDIT

Fig. 11 presents additional experimental results of our method, demonstrating that the proposed $D^2$-Edit not only enables virtual try-on and original fashion image editing but also allows simultaneous editing of the try-on clothing. These edits encompass various aspects, including *color, sleeve length, material, texture, clothing type*, etc.

## C EXPERIMENTAL COMPARISON WITH SOTA METHODS

### C.1 DETAILS OF BASELINES

To verify the effectiveness of $D^2$-Edit, we compare it with the following baselines:
**Break-A-Scene Avrahami et al. (2023) & blended diffusion Avrahami et al. (2022) (BAS-BD)**, where BAS is a method for extracting multiple concepts from a single image, allowing flexible

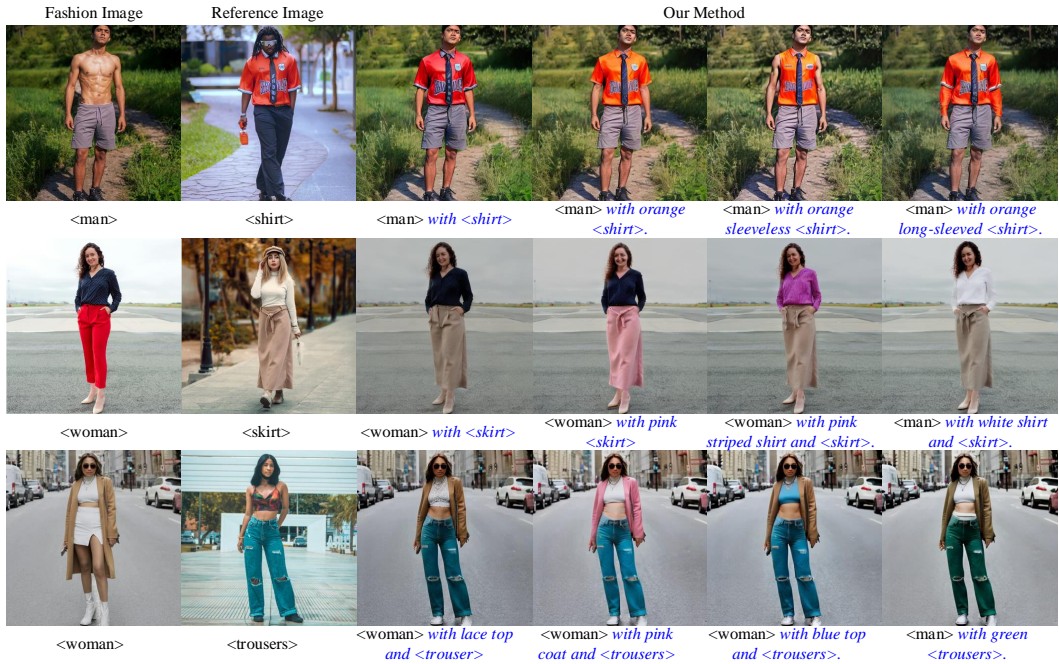

Figure 11: More results of our method. Our method enables garment try-on, original fashion image editing, and the simultaneous editing of try-on images. Blue colored texts indicate added or modified garment concepts.

editing of specific concepts using text prompts. BD is an image generation method combining the diffusion model with blended latent space, which generates high-quality image variants by mixing different concepts in the latent space. This method can create diverse and realistic image variants by manipulating the underlying latent representations. By combining BAS with BD, the personalized editing of fashion images can be achieved.

**Anydoor Chen et al. (2024)** is a method that utilizes both the source and reference images and their respective target masks to guide image editing. This method allows the integration of objects from the reference image into the original image. It combines advanced text-driven image editing methods such as Turbo-Edit Deutch et al. (2024), LEDITS++ Brack et al. (2024), and T-FIT Huang et al. (2025) to achieve even more diverse editing results.

**Mimicbrush Zhao (2024)** is a reference image-based editing method that enables fashion image editing by a reference image and a user-supplied mask. It can also be combined with text-driven methods to realize fashion image editing.

**TexFit Wang & Ye (2024)** is a text-driven fashion image editing method designed to enable local editing of fashion images using textual guidance.

**ACE++ Mao et al. (2025)** is an instruction-based diffusion framework that tackles various image generation and editing tasks, which follows a two-stage training scheme.

## C.2 COMPARISON WITH TEXT-DRIVEN IMAGE EDITING METHODS

To compare the performance of our method with text-driven image editing methods, we conduct experiments from two perspectives: original fashion image attribute editing and virtual try-on. On one hand, we directly compare various methods for editing attributes such as *fabric, color,* and *sleeve length* in the source image; on the other hand, we evaluate text-driven methods versus our method in the try-on task by providing detailed text prompts for the clothing. Therefore, we choose three state-of-the-art text-driven image editing methods, i.e., LEDITS++ Brack et al. (2024), Turbo-Edit Deutch et al. (2024), and T-FIT Huang et al. (2025) as baseline. The experimental results, presented in Fig. 13 and Fig. 12, reveal the following:

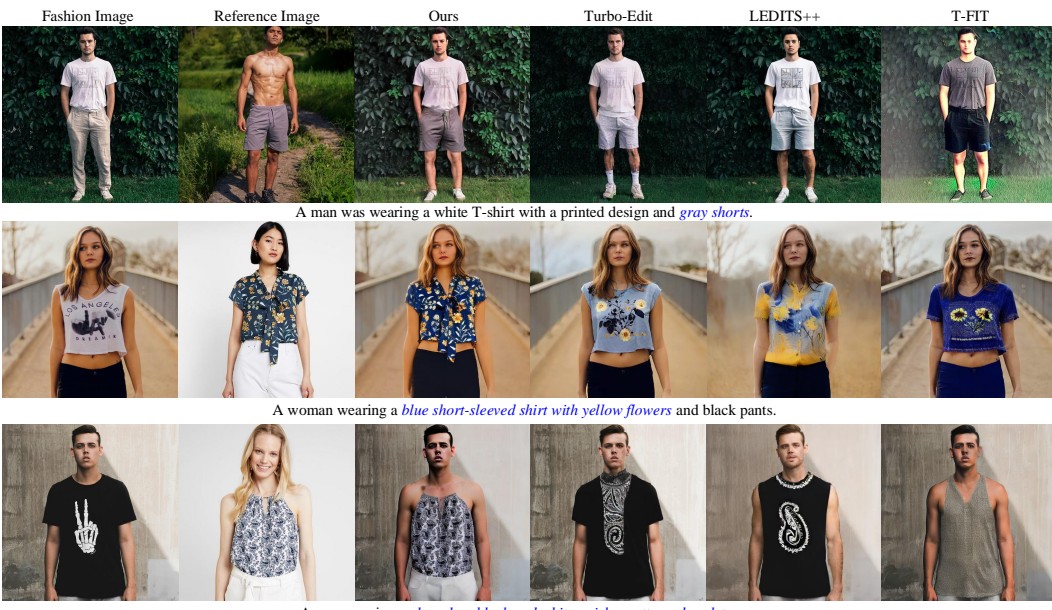

Figure 12: Experimental comparison with a text-driven image editing methods on virtual try-on task. Blue text is used to label edited garment concepts or attributes.

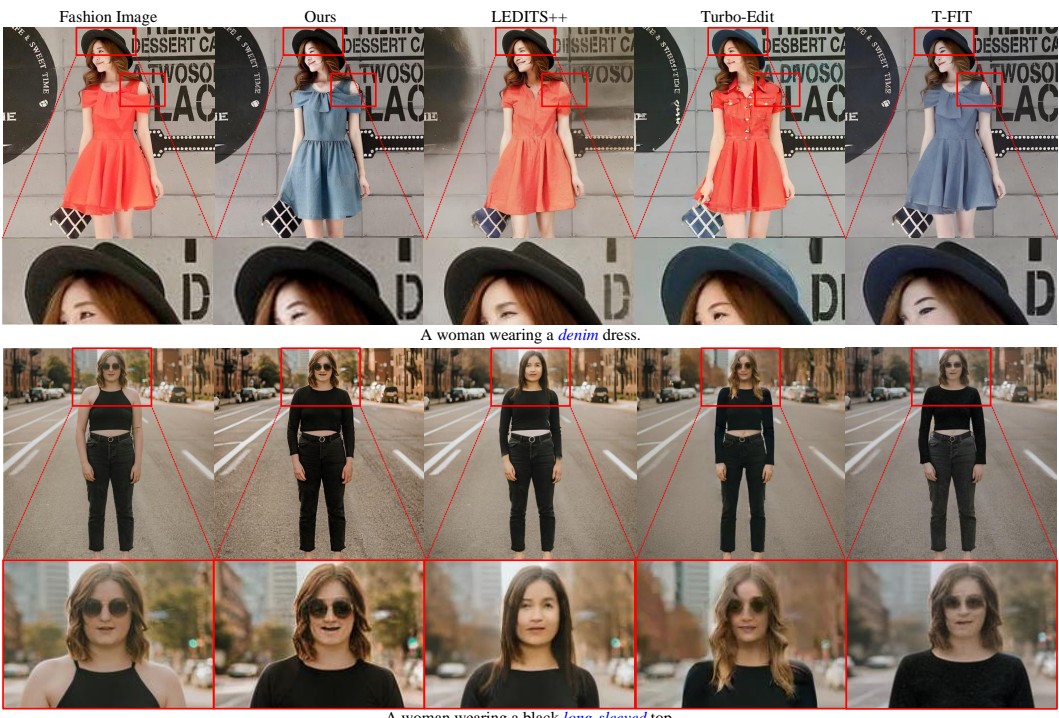

Figure 13: Experimental comparison with text-driven image editing methods on the original fashion image editing task. Blue text is used to label edited garment concepts or attributes; red boxes are used to highlight significant differences in the edited image.

(i) In the virtual try-on task, even when detailed text descriptions are provided, text-driven methods fail to fully reproduce the try-on effect, primarily because text alone is insufficient to capture the intricate details of clothing (see Fig. 14). Conversely, our method achieves promising results in virtual try-on while keeping unrelated regions unaffected.

(ii) In the original fashion image attribute editing task, text-driven image editing methods often induce unintended changes to unrelated regions. For example, as shown in the first row of Fig. 13, LEDITS++ Brack et al. (2024) alters the shoulder details of a skirt and severely disrupts the background, while Turbo-Edit Deutch et al. (2024) erroneously modifies the skirt's style—likely due to dataset biases—and may even change the subject's identity. Although T-FIT Huang et al. (2025) achieves competitive results in clothing attribute editing, it still struggles to preserve the background and often unintentionally alters non-target regions when editing a specific clothing attribute. For example, as shown in the Fig. 13, when editing the skirt fabric to denim, the material of the hat is incorrectly changed. In contrast, our method preserves both background and identity while achieving superior visual quality, demonstrating more precise and fine-grained editing of fashion attributes.

(iii) Table 3 presents the quantitative performance of our method compared to text-driven methods on the fashion image editing task. As shown, our method achieves the best results in both CLIP-I and CLIP-T. This indicates that the edited images produced by our method not only align well with the given text prompts but also preserve irrelevant regions of the image, ensuring more faithful editing.

Overall, our method outperforms existing text-driven methods in both virtual try-on and original image fashion attribute editing.

### C.3 Comparison with Virtual Try-on Methods

We also compare our proposed method with state-of-the-art virtual try-on methods, including VITON-HD Choi et al. (2021), DCI-VTON Gou & Sun (2023), and StableVITON Kim et al. (2024). As shown in Table 3, our method demonstrates competitive performance on the virtual try-on task.

Table 3: Performance comparison with the image editing & try-on methods

| Category | Methods | CLIP-I ↑ | CLIP-T ↑ |
|---|---|---|---|
| **Fashion Image Editing** | Turbo-edit (SIGGRAPH Asia 2024) | 0.9101 | 0.1547 |
| | UltraEdit (NeurIPS 2024) | 0.9056 | 0.1651 |
| | LEDITS++ (CVPR 2024) | 0.9160 | 0.1367 |
| | G & R (ECCV 2024) | 0.9116 | 0.1529 |
| | T-FIT (CVPR 2025) | 0.8878 | **0.2118** |
| **Virtual Try-on** | StableVITON (CVPR2024) | 0.8362 | 0.1329 |
| | DCI-VTON (ACM MM 2023) | 0.7520 | 0.1344 |
| | VITON-HD (CVPR 2021) | 0.7402 | 0.1226 |
| **Both** | Our (Only edit) | **0.9561** | 0.1558 |
| | Our (try-on & edit) | 0.9386 | 0.1656 |

## D   Sensitivity Analysis

To evaluate the sensitivity of our model to key weight parameters, we conduct experiments to examine the impact of the degradation weight $\alpha$ and the concept intensity modulation weight $\gamma$ on the editing results.

Fig. 14 displays the outputs under different degradation weight $\alpha$. When $\alpha$ is low, the edited images tend to blend features from the reference image—for instance, causing the subject's body to appear bulkier and their position to shift. As $\alpha$ increases, the generated images progressively preserve more information from the original fashion image and are less influenced by the reference image. Notably, when $\alpha$ is set to 1, the image achieves the optimal performance, retaining irrelevant attributes while effectively implementing virtual try-on.

The effects of varying the concept intensity modulation weights $\gamma$ on image attribute modifications are also illustrated in Fig. 14. As $\gamma$ increases, the intensity of the edited attribute changes gradually. For example, given the text prompt "a <man >with green <sleeveless top >", the garment's color

Figure 14: Experimental results under varying degradation weight $\alpha$ and concept intensity modulation weight $\gamma$ settings. Row 1 shows the editing outcomes using the text prompt 'a <man >with <sleeveless top >' under different degradation weights. Row 2 presents the editing outcomes using the text prompt 'a <man >with green <sleeveless top >' under different concept intensity modulation weights.

shifts through progressively stronger shades of green, thereby offering enhanced controllability for fashion image editing.

# E    QUANTITATIVE RESULTS OF THE ABLATION STUDIES

We present quantitative results (of Fig. 5) in the Table 4, confirming the positive contribution of each component. Note that, when the CAEDIM module is ablated from our method, the model struggles to produce significant structural modifications (e.g., adjusting *sleeve length*), resulting in edited images that remain highly similar to the originals. Consequently, although the model achieves strong performance on LPIPS, PSNR, and CLIP-I, it fails to effectively follow text instructions for structural editing, leading to inferior CLIP-T scores. This highlights the critical role of the CAEDIM module in fashion image editing, as it enables flexible and accurate semantic-guided attribute manipulation.

Table 4: The quantitative evaluation of the ablation study

| Model | LPIPS ↓ | PSNR ↑ | CLIP-T ↑ | CLIP-I ↑ |
|---|---|---|---|---|
| w/o CAEDIM | 0.2560 | 0.2339 | 0.1137 | 0.9770 |
| w/o $\mathcal{L}_{\text{masked\_diff}}$ | 0.4042 | 0.1277 | 0.1343 | 0.9517 |
| w/o IDM | 0.4669 | 0.0825 | 0.1447 | 0.9554 |
| Our (Fig. 5, row 1) | 0.3525 | 0.1806 | 0.1369 | 0.9593 |
| Our (Fig. 5, row 2) | 0.3281 | 0.1889 | 0.1481 | 0.9554 |

# F    RELATED WORK

**Text-driven Fashion Image Editing.** Text-driven fashion image editing has made significant progress in recent years, enabling the precise editing of fashion images based on a given textual description. These methods have evolved from GAN-based methods Zhu et al. (2017); Jiang et al. (2022); Pernuš et al. (2025) to diffusion models Baldrati et al. (2023; 2024); Wang & Ye (2024); Anonymous (2024). TexFit Wang & Ye (2024) localizes editing regions using only text, while DPDEdit Anonymous (2024) enhances precision by integrating Grounded-SAM Ren et al. (2024) and leveraging multimodal inputs. However, even with the aid of multimodal information, accurately describing the details of the target garment remains challenging, thereby hindering the achievement of virtual try-on.

**Image-based Virtual Try-On.** Image-based virtual try-on method aims to generate the target image

sharing the same identity as the input portrait in a fashion image while wearing the specific garment Lin et al. (2023); Zhu et al. (2023); Zeng et al. (2024); Cui et al. (2024). Most methods are trained on paired datasets, like VITONHD Choi et al. (2021) and DressCode Morelli et al. (2022), which can achieve high-quality results on in-domain images. However, they struggle with generalizing to out-of-domain data and cannot be trained when paired data is unavailable. Moreover, these methods often rely on a well-designed fashion product image. Therefore, several works Xie et al. (2021); Cui et al. (2024) are designed to exchange garments between two street-style images with different portraits, without requiring a product reference image. Despite this progress, these methods are limited to virtual try-on and cannot perform fine-grained attribute editing on the generated images, e.g., altering the color or fabric of the clothing. In our work, the proposed method enables simultaneous fashion image editing and virtual try-on in an end-to-end manner.

**Image Personalization.** Image personalization aims to identify a personalized concept from user-provided images and guide the generation of new images containing the learned concept Avrahami et al. (2023); Safaee et al. (2024); Zhou et al. (2024). Initial approaches such as textual inversion Gal et al. and DreamBooth Ruiz et al. (2023), addressed this task by either optimizing a text embedding or fine-tuning the entire T2I model. Additionally, the research community has widely adopted low-rank adaptation (LoRA) Hu et al. (2021) for personalization, offering an efficient and lightweight solution. Besides, numerous works Ding et al. (2024); Woo & Kim (2025) have explored tuning-free approaches to personalization. However, these methods often rely on training an encoder with extensive domain-specific image datasets. In contrast, our method can achieve precise fashion image personalization (i.e., editing and virtual try-on) with just a reference image, a source image, and the corresponding target text prompts.

# G LIMITATIONS

Although our method demonstrates robust performance in real-world fashion image editing and supports background modification—a promising feature for enhancing style transfer—it does have certain limitations. Specifically, when processing images with solid backgrounds—especially white—the method encounters challenges, resulting in edited images that tend to exhibit a darker tone, as shown in Fig. 15. This issue represents a significant challenge that requires further investigation. To mitigate this problem, we plan to incorporate perceptual color spaces, like CIELAB, to overcome this limitation.

| Fashion Image | Reference Image | Ours | |
|---|---|---|---|

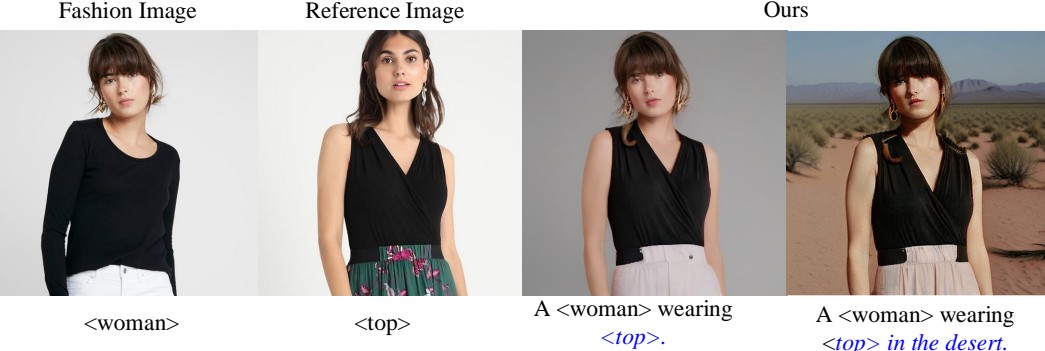

| | | A <woman> wearing *<top>*. | A <woman> wearing *<top> in the desert.* |
|---|---|---|---|

Figure 15: The limitation of our method.

