# OpenReview forum: "Joint Learning Between Reference Image and Text Prompt for Fashion Image Editing"
_ICLR.cc/2026/Conference — ICLR 2026 Conference Withdrawn Submission_

### Official Review · Reviewer_aY8G · 2025-10-28

**Soundness:** 2
**Presentation:** 2
**Contribution:** 2
**Rating:** 4
**Confidence:** 2

**Summary:**

This paper proposes $D^2$-Edit, a single-stage framework that combines a reference image and a text prompt to perform virtual try-on and attribute editing (e.g., color, material, sleeve length). The method is evaluated on the StreetTryOn and Unsplash datasets, reporting whole-image quality, garment-region metrics, and inference time.

**Strengths:**

- A single-stage pipeline unifying reference-guided shape/texture transfer with text-driven attribute control, motivated by a clear discussion of error accumulation and inefficiency in two-stage methods.
- Lightweight training: LoRA fine-tuning only, on a single RTX 4090 GPU.
- The problem setting—dressing with a reference image and refining via language—is practically meaningful.

**Weaknesses:**

- The code is not provided, resulting in low reproducibility.
- The proposed model heavily relies on masks generated by Grounded-SAM. However, there is no robustness analysis against mask errors, nor any evaluation of the model’s behavior without reference masks (or with coarse masks).
- While the paper reports the accuracy for the entire image and the clothing region, it lacks separate metrics for the background and face, leaving it unclear whether those regions are preserved properly.
- Important hyperparameters $\gamma, w, \alpha$, the specific list of attribute pairs $t_0, t_1$, and the templates of prompts $P_s, P_r$ used for training are insufficiently described, making reproduction impossible for readers.
- There remain several issues in notation and presentation. Please refer to the following points and revise where applicable.

**Questions:**

- Although the method is designed for joint text+reference inputs, please report text-only and reference-only ablations to quantify each modality’s contribution. A common workflow could first involve dressing with the reference and then applying small text edits; such ablations would clarify feasibility and user guidance.
- While successful cases are presented, could you also report and discuss some failure cases?
- Many other fashion generation models have been proposed—are the methods chosen as baselines appropriate and representative?

(Other comments)
- General: Add parentheses to citations where needed.
- Figure 1: In (b) and (d), some output images overlap; the layout looks unnatural. The “Generative Model” label also appears off-center.
- Figure 2: Include concrete prompt examples beyond \(t_0, t_1\) and “A photo of \<skirt\>/\<woman\>.” “Base Image” may read clearer than “Fashion Image.”
- Notation: Consistency is key; if possible, use bold lowercase for vectors, or otherwise ensure a consistent style.
- Line-level fixes: clarify relationships among $v$, $t_0, t_1$, $P_s$, $P$, $P_r$; “2e-2,,” -> “2e-2,”; italicize subscripts in $e_t, e_\phi$; $s_{\rm tar}$ -> $s_{tar}$; avoid overloading $t$ for both time and tokens (e.g., rename tokens to $v_0, v_1\$.
- Figure 4 (bottom): The images in “Ours” are misaligned; please tidy the layout.
- Spelling: “Morerover” → “Moreover.”

---

### Official Review · Reviewer_yr8E · 2025-10-31

**Soundness:** 2
**Presentation:** 3
**Contribution:** 2
**Rating:** 4
**Confidence:** 2

**Summary:**

This paper introduces $D^2$-Edit, a framework for fashion image editing that jointly learns from a reference image and a text prompt. The key problem being addressed is that existing methods are typically specialized: image-based virtual try-on methods can transfer a garment but cannot edit its attributes, while text-based editing methods can alter attributes but lack the precision to perform a faithful virtual try-on from a reference photo. The proposed $D^2$-Edit framework aims to achieve both simultaneously in a single, end-to-end model. It uses (1) an Image Degradation Module to isolate the garment concept in the reference image, (2) an Image Reconstruction Module with a masked diffusion loss to learn to reconstruct the target garment, (3) a Garment Concept Learning Module to bind this visual concept to a text token via cross-attention, and (4) a Concept Editing Direction Identification Module (CAEDIM) at inference time to apply fine-grained attribute edits specified by text.

**Strengths:**

The paper tackles a clear and valuable problem in generative image editing. The goal of combining the "best of both worlds" (reference-based garment transfer and text-based attribute modification) is highly practical for fashion e-commerce and design. The proposed method successfully avoids the error accumulation and high latency of a naive two-stage approach (try-on first, then edit).

**Weaknesses:**

1. The Image Degradation Module (IDM) and the masked loss ($\mathcal{L}_{masked-diff}$) are both critically dependent on the quality of the segmentation mask ($M_c$) generated by Grounded-SAM. The paper does not analyze the model's robustness to segmentation failures. If Grounded-SAM produces an inaccurate mask (e.g., due to occlusion, complex patterns, or challenging poses in the reference image), it seems likely that the garment concept learning would be corrupted, leading to poor results.

2. The CAEDIM module is shown to work for color, material ("leather"), and length. It is less clear how well it would perform on more complex structural changes (e.g., "change from a v-neck to a crew-neck") or highly intricate patterns. The "leather" example in Figure 4 (row 2) appears to lose some of the original pleating from the reference skirt.

**Questions:**

1. Could the authors comment on the model's performance when the Grounded-SAM mask ($M_c$) is imperfect? How sensitive is the framework to noise or inaccuracies in this initial segmentation step?
2. Have the authors tested its ability to apply complex patterns (e.g., "add argyle pattern") or significant structural edits (e.g., "add a hood," "remove the pockets")?

---

### Official Review · Reviewer_UBu5 · 2025-10-31

**Soundness:** 2
**Presentation:** 1
**Contribution:** 2
**Rating:** 2
**Confidence:** 3

**Summary:**

The task of multi-modal image editing is very useful, especially with fashion images. There is novelty in some proposed components of the work, particularly Attribute Editing Direction Identification (Section 2.6), but the overall presentation of the paper is not easy to follow in its current form -- key notations are inconsistent across different equations and across training and inference, making it hard unclear about the input and output of the model and what it actually does. Noticeable artifacts are present in the final result, with loss of human identity and over-saturation.

**Strengths:**

This work tackles a practical and useful problem: multi-modal virtual try-on that takes in multiple images and texts as input, and output the edited image. While further details regarding the proposed Attribute Editing Direction (Section 2.6) could be clarified, the idea seems to be a novel proof of concept, despite that over-saturated artifacts exist.

**Weaknesses:**

1) Inconsistent notation making it harder to understand what the model does:
- In Eqn2 and Eqn3, it seems like separate noises are sampled for source image and reference image, and the diffusion model uses separate forward passes to denoise source image and reference image. But in algorithm 1 training phase, it reads like a shared noise is used across source and reference images, and the diffusion model takes in source and reference images simultaneously in a single forward pass. Finally, in algorithm 1 inference phase, there is no reference image notation. All together, it is hard to follow exactly what is the model's input and output.
- Eqn6 has only hyper-paramter \lambda, while the text only describes hyper-parameters \gamma and \textit{w}

2) Are there more concrete examples for what values P, Ps, t0, t1 are?

3) Artifacts are rather noticeable in the result (e.g., Figure 3):
- human's identity changed
- over-saturated colors

**Questions:**

Is the over-saturation caused by the proposed attribute editing direction formulation (Section 2.6)? In Figure 5, w/o CAEDIM has noticeably less over-saturated (or say, intensely contrasted) result compared to the full model.

Also, it seems to me that the attribute editing direction formulation bears similarity to Adaptive Projected Guidance [1], and that work found that mainly the orthogonal direction helps guiding while the parallel direction contributes to over-saturating. Wonder if the authors have explored the APG idea to see if that alleviates the over-saturating effect?

[1] "ELIMINATING OVERSATURATION AND ARTIFACTS OF HIGH GUIDANCE SCALES IN DIFFUSION MODELS", Seyedmorteza Sadat, Otmar Hilliges, Romann M. Weber, ICLR 2025

---

### Official Review · Reviewer_YhdT · 2025-11-02

**Soundness:** 3
**Presentation:** 2
**Contribution:** 3
**Rating:** 6
**Confidence:** 2

**Summary:**

This paper presents a framework that enables garment migration and attribute editing via textual descriptions given a reference image. There are four main components: (1) image degradation module, (2) image reconstruction module, (3) garment concept learning module, and (4) concept editing direction identification module.

**Strengths:**

- The application of this method is intuitive, easy, and useful for real world use. There is no need to provide auxiliary information such as a segmentation mask / skeleton image... A reference image and a text prompt are sufficient for virtual try on and fashion editing.

- The results are very strong.

- At first, I was unsure how this method differs from previous methods, but Fig. 1 helped me understand it clearly.

**Weaknesses:**

- In Fig. 2, it would be helpful to include a high level explanation of what each module does and how the data flows. Also, in the predictions, should the final edited image be presented (that is, the subject in the fashion image wearing the skirt from the reference image)? I understand this is to explain image reconstruction, but it may be better to present the training and inference figures separately. In addition, attribute editing direction identification is not presented in Fig. 2. Currently, it is a bit misleading, and readers might be confused since there appears to be no change.

- It is not very clear how the fashion editing score can be computed and how supervision can be done given the absence of ground truth with edited attributes. I am not familiar with this domain, so please provide a detailed explanation for readers like me.

**Questions:**

- Provide more details on fashion attribute editing supervision using the attribute score.
- Revise Fig. 2 to be more informative, for example by providing separate flow figures for training and evaluation.

---

### Note · Authors · 2025-11-14

I have read and agree with the venue's withdrawal policy on behalf of myself and my co-authors.